# Multi-Criteria Evaluation of Best Available Treatment Technology for Waste Lead-Acid Battery: The Case of China

**Wei Wang [1], Yi He [2], Deyuan Zhang [3], Yufeng Wu [1,\*] and Dean Pan [1]**

[1] College of Materials Science and Engineering, Beijing University of Technology, Beijing 100124, China; weiwei3r@163.com (W.W.); pandean@bjut.edu.cn (D.P.)

[2] Solid Waste and Chemicals Management Center, Ministry of Ecology and Environment, Beijing 100029, China; heyi@mepscc.cn

[3] Institute of Economic System and Management, National Development and Reform Commission, Beijing 100035, China; zhangdy271@126.com

\* Correspondence: wuyufeng3r@126.com; Tel.: +86-10-6739-6234

**Abstract:** Improper waste lead-acid battery (LAB) disposal not only damages the environment, but also leads to potential safety hazards. Given that waste best available treatment technology (BATT) plays a major role in environmental protection, pertinent research has largely focused on evaluating typical recycling technologies and recommending the BATT for waste LABs. First the evaluation indicators were selected based on the analysis of main factors affecting the pollution control of waste LAB treatment. The relative weights of each indicator were determined via the Delphi-attribute hierarchy model (AHM) in the second step. To determine the BATT, the attributive mathematics theory was adopted to calculate the attribute measure of single and multiple indices. Then, five recycling technologies commonly used in the secondary lead industry were estimated using the proposed evaluation system, and the feasibility of the recommended BATT was preliminarily verified. The results indicated that mixed smelting technology (MST), pre-desulfurization and multi-chamber smelting technology (PD-MCST), and direct smelting technology (DST) were found to perform well and were therefore deemed optimal for waste LAB disposal at this stage. The validation study showed that the DST can meet the requirements of pollution control, which is consistent with the evaluation results.

**Keywords:** waste lead-acid battery; best available treatment technology; evaluation; attribute hierarchy model; attribute mathematics theory

## 1. Introduction

After more than 150 years of continuous development and improvement, lead-acid batteries (LABs) have become a widely used chemical power source worldwide, with good electrochemical reversibility, stable voltage characteristics, and wide application range [1–3]. According to 2017 statistics, the global market share for LABs in the rechargeable battery industry was still over 70%, reaching USD 42.9 billion [4]. Moreover, LAB consumption is expected to steadily grow as the automobile, electric bicycle, and energy storage industries continue to expand [5]. The average service life of a LAB is approximately two years. Large-scale production would inevitably lead to an increase in the amount of used LABs, making the recycling of waste batteries an industrial problem. Improper waste LAB disposal not only damages the environment, but also leads to potential safety hazards. Lead and its compounds are a kind of nondegradable pollutant, and its properties are also relatively stable [6]. Generally, these can flow into the environment through wastewater, waste gas, or waste residue, which

can cause severe pollution threats and health problems. According to the World Health Organization (WHO), approximately 1.2 billion people all over the world live in a lead-polluted environment, and about 99% of these occur in non-developed countries. Peng et al. found that there is a close relationship between secondary lead enterprises and lead pollution [7].

For these reasons, all lead smelting operations must be regulated by the government in some way. Developed countries have very comprehensive laws and regulations regarding pollution control and best practices for waste LAB treatment. These regulations include the Clean Air Act, Clean Water Act and Mercury-Containing and Rechargeable Battery Management Act in the United States; the Waste Battery Management Law in Germany; and the Resource Recycling Law in Japan. China has also issued the Technical Specifications of Pollution Control for Treatment of Waste Lead-acid Batteries. These laws and regulations not only require companies to adopt advanced technology to reuse lead grid, lead paste, plastic, battery separators, and electrolytes separately under closed conditions and negative pressure, but also to specify the final emission limits of various pollutants [8]. However, the guidelines of said measures to determine the BATT for pollutant control are too general and lack scientific rigor. Some technical, economic, environmental, resource, and energy constraints have made the evaluation a multi-criteria decision-making problem. In order to solve such a complex problem, we must fully consider both qualitative and quantitative aspects. Some scholars have conducted preliminary research. Tian et al. described a comparative study of five typical LAB recycling processes by compiling data about the input materials, energy consumptions, pollution emissions, and final products. They found that not all of the innovative hydrometallurgical processes are healthy alternatives, and attention should be paid to indirect emissions in the environmental inspection [9]. Genaidy et al. established strategies to increase lead recovery, prevent pollution, and minimize waste via a systematic review and critical appraisal of the published literature. They proposed that the adoption of cleaner technologies at the preprocessing stage in secondary smelter operations can significantly improve the smelter performance from both economic and environmental perspectives [10]. Li et al. accounted for all the integrated assessment indicators for three typical secondary lead smelting technologies using substance flow analysis. They indicated that lead pollution emission load is the result of co-control of process pollution prevention and end-of-pipe control, and the hydrometallurgical smelting process will be the best available smelting technology for the secondary smelting industry [11].

However, the abovementioned studies have their limitations. Specifically, they either fail to consider the entire process and limit the system boundaries from lead paste to lead product, or fail to fully consider other major pollutants in addition to those containing lead. There are two stages in the recovery of waste LABs: physical separation (breaking and separation) and chemical separation (smelting and refining). Each process involved in the two stages will produce different pollutants. It was proved that relying only on the traditional end-of-pipe treatment cannot effectively reduce pollution, and the cost of operation and maintenance is high [12]. There is a need to replace obsolete technology with cleaner alternatives (i.e., technology and equipment with low emissions, saving resources and energy, and economic feasibility, etc.). Therefore, it is urgent and necessary to develop a scientific and systematic evaluation system for the evaluation of BATT for pollutant control during the entire process. This paper establishes the evaluation system of the BATT for waste LABs by the field survey, literature, and expert seminars. In order to take full advantage of experiences and reduce subjective randomness, the relative weights of indicators are determined via Delphi-attribute hierarchy model (AHM). Given that attribute mathematics theory can successfully resolve the measurement problem of qualitative description, the relationship between different qualitative descriptions, and the relationship between corresponding measurements in comprehensive evaluations, this study seeks to evaluate the BATT via attribute mathematics by setting evaluation criteria, and the pollution control level of the recommended BATT is preliminarily verified.

## 2. Literature Review

At present, the evaluation system of the BATT for waste LABs has not formed a consistent standard, but it must be able to fully reflect the status of each treatment technology. Evaluation of the BATT is an integrated problem combining technology, economy, environment, resource, and energy. Each technology has different characteristics, so the selection of evaluation indicators should follow the principles of systematization, typicality, dynamics, independence, and operability. To identify criteria and sub-criteria for BATT evaluation, the related research in the field of battery recycling was reviewed.

Tian et al. studied five LAB recycling processes based on life-cycle assessment and found that the QSL furnace is the best choice. They also pointed out that the indirect environmental impacts rely on the consumption of materials and energy [9]. The energy needed to provide heat depends on specific methods, including oil, gas, coke, electricity, etc. There are also several different types of equipment in which the smelting process may be carried out: reverberatory furnace, blast furnace, rotary kiln, and QSL furnace, etc. Different energy sources and equipment have different environmental impacts. Obviously, smelting equipment and processes, material and energy consumption should pay more attention in the BATT evaluation. Peng et al. analyzed the potential lead pollution during the pyrometallurgical process from four aspects, including atmosphere, soil, water, and human exposure. They realized that smelting is the main process to produce $SO_2$ and lead dust, and the pollution control levels of different smelting furnace types are different. The main sources of lead pollution in the soil were the random dumping of waste residue and the dust produced in the smelting process [13]. In other words, optimizing the smelting process and strengthening environmental management (i.e., environmental management of the production process, solid waste management, etc.) will contribute to pollution control. Faé Gomes et al. attempted to reduce waste generation through improvements in the process and material inputs and modify hazardous slag compounds. They indicated that the amount of waste slag is related to the type of furnace used [14]. Only waste minimization and reduction of slag toxicity can lessen the overall environmental impact of the process. Bourson not only introduced the process and main work processes, but also fully considered the economic aspect. He believed that the wastes generated by the process can be either reused or eliminated, and the process itself is profitable. He suggested realizing the comprehensive resource utilization on the premise of safety [15]. Eckel et al. investigated 12 suspected lead contamination sites and found that 10 of them were former secondary lead smelters or lead works. It was clear that the construction of the disposal site and the dust emission control will contribute to the prevention of soil pollution in the future [16]. Although the disposal site is not a key indicator of environmental monitoring, it will lead to long-term potential pollution. Once an accident occurs, it will eventually attract people's attention. Kimbrough and Carder found that facilities with air emission problems also had water discharge problems. The source of these problems was identical to the air emission problems, namely, little direct point source emissions and mainly fugitive emissions from improper storage and transport of feedstock and hazardous waste [17]. We have reason to believe that disposal site and internal environmental management will have a significant impact on potential pollution. As a common problem in the industry, a large amount of discard slag is produced in the smelting process. Considering the environmental (discard slag represents a main source of pollution) and economical (high disposal cost) effects of slag management, Angelis et al. believed that the slag must be reused finally. The high release of lead from the solidification products seems to be a limiting factor for a reusable material and must be limited. They also realized that stabilizing the slag ultimately means controlling the fusion–reduction–refining process, which will be a huge step towards cleaner production, as this is the main hazardous waste formation in the entire process [18]. Kreusch et al. analyzed the main sources of environmental impact caused by the stages of the recycling process, including acid electrolyte, particulate lead, lead-contaminated scraps, lead-contaminated dust, $SO_2$, production of slag. They proposed process improvements aimed primarily at increasing production output by reducing the loss of lead in slag and particulates, thereby providing a healthier work environment [19]. Gottesfeld et al. recommended that comprehensive industry-specific regulations be in place, including performance measures for stack emissions, ambient

air, occupational exposure levels, minimum production capacity for new and existing recycling plants, and waste disposal. The results showed that the comprehensive laws and regulations can effectively control lead pollution [20]. Bicanová et al. counted the lead outputs into the environment through the leaks of this contaminant into the atmosphere, water, and soil, and through its transfer in wastewater and waste [21]. In this way we believe that it is very important to improve the recovery rate of lead in the whole process. Rajčević et al. evaluated the blood lead levels in children living in two villages in Serbia, and indicated a contribution of 25%–40% of the take-home lead exposure in the blood lead levels of children living in the vicinity of a secondary lead smelter [22]. So attention should be paid to environmental management, lead dust emission, and wastewater in smelters.

The literature review in the above context revealed that, although there are in-depth analyses and demonstrations of certain factors affecting secondary pollution, there is no comprehensive framework to evaluate the pollution control for waste LAB treatment. Thus, to promote the BATT evaluation, we identified the indicators that affect the pollution control of waste LAB disposal through a literature review. Finally, the identified criteria were divided into six dimensions: environmental effect, comprehensive resource utilization, technical performance, material and energy consumption, economic performance and environmental management. Each indictor and its definitions are summarized in Table 1.

**Table 1.** Evaluation system of the best available treatment technology (BATT) for waste lead-acid batteries (LABs).

| Criteria | Sub-Criteria | Definition | Reference |
|---|---|---|---|
| Environmental effect (A1) | Lead dust emission (A11) | Proportion of Pd dust emission to Pd production. | [6,17,19,22] |
| | Lead content of discard slag (A12) | Pb content in discard slag. | [14,18] |
| | $SO_2$ emission (A13) | Proportion of $SO_2$ emission to Pd production. | [19,23] |
| | Discard slag (A14) | Proportion of discard slag to Pd production. | [6,13–15,18,23] |
| | NOx emission (A15) | Proportion of NOx emission to Pb production. | [23] |
| Comprehensive resource utilization (A2) | Utilization of lead (A21) | Proportion of Pb content in products versus that of original waste LAB. | [21,22] |
| | Utilization of sulfur (A22) | Utilization rate of sulfur in waste LAB through comprehensive utilization in various ways. | [19] |
| | Disposal of electrolyte (A23) | Proportion of safety treated electrolyte versus total electrolyte. | [17] |
| | Disposal of discard slag (A24) | Proportion of safety treated discard slag versus total discard slag. | [13,14] |
| | Utilization of plastic (A25) | Proportion of recycled plastic versus the original plastic in waste LAB. | [15] |
| | Utilization of wastewater (A26) | Proportion of reused wastewater versus total wastewater. | [6,15,17,24] |
| Technicalperformance (A3) | Industrial policies (A31) | Degree of policy compliance to encourage industrial development (e.g., Promotion plan for the development of the regenerated nonferrous metal industry, Technical policy for pollution prevention and control in lead-acid battery production and its regeneration process, Specification for the secondary lead industry, etc.). | [20,24] |
| | Smelting process and equipment (A32) | Process and equipment used in smelting process (e.g., Oxygen-enriched smelting, Pure oxygen smelting, Continuous lead-melting furnace, Closed melting furnace, etc.). | [3] |
| | State of technology reliability (A33) | Ensuring technology works consistently and reliably. | [25] |
| | Automation level (A34) | Automation level of process and equipment in the entire process (e.g., Automatic crushing and separation, Automatic feeding system, Automatic slag removal, Automatic monitoring, etc.). | [25] |
| | State of disposal site (A35) | Construction of disposal site (e.g., closed, negative pressure and anti-leakage, etc.). | [13,16] |
| Material and energy consumption (A4) | Comprehensive energy consumption (A41) | Total energy consumption in the entire process. | [9,25] |
| | Fresh water consumption (A42) | Total freshwater consumption in the entire process. | [5] |
| | Auxiliary materials consumption (A43) | Total auxiliary material consumption in the entire process. | [5,25] |
| Economic performance (A5) | Investment profit (A51) | Proportion of profit versus total investment. | [15,25] |
| | Operational cost (A52) | Operation costs associated with the treatment of one ton of waste LABs. | [15,24,25] |

**Table 1.** *Cont.*

| Criteria | Sub-Criteria | Definition | Reference |
|---|---|---|---|
| Environmental management (A6) | Environmental laws and regulations (A61) | Degree of law and regulation compliance to encourage environmental protection (e.g., Clean production standard–waste lead acid battery recycling industry, Technical specifications of pollution control for the treatment of lead-acid batteries, Emission standards of pollutants for the secondary copper, aluminum, lead and zinc industry, etc.). | [20,24] |
| | Environmental management system (A62) | Environmental management system, organization and professionals (e.g., Control requirements for waste gas, wastewater and solid waste, Operating procedures for production processes, Quality inspection system for raw materials and auxiliary materials, Energy consumption quota management system, etc.). | [13,20,24] |
| | Environmental emergency (A63) | Necessary measures for the environmental pollution accident (e.g., Emergency preparedness and response system of environmental pollution accident). | [7,26] |

## 3. Methodology

### 3.1. AHM Method

The attribute hierarchy model (AHM) is an unstructured decision-making method developed on the basis of analytic hierarchy process (AHP) theory [27–29]. Compared with the AHP, the AHM is based on game models and does not require consistency examination because of the underlying pairwise comparisons within the measure matrix [30]. It has a mature theoretical basis and practical experience and can be widely used in the field of technology evaluation [29,31]. Ma et al. developed a systematic post project evaluation index and determined weights for all indicators by using the AHM [32]. Hemalatha et al. determined the relative weights of the service quality dimensions and their enablers through AHM to avoid consistency check, present in the standard AHP method [30]. Qiang et al. evaluated the energy demand and environmental impacts, as well as the water requirement of the polylactide-based wood plastic composites during the cradle-to-gate stages based on life-cycle assessment. They also used the AHM to determine the weighting factors of the different environmental impact categories to the environmental impact load (EIL) [33]. So this paper applied the AHM to the quantitative and qualitative calculation of each index weight, and the specific steps were as follows (see Figure 1).

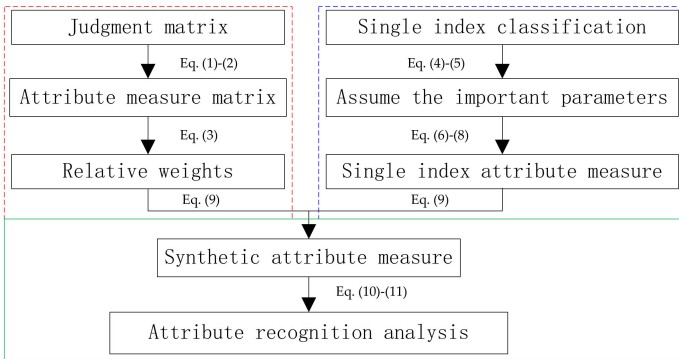

**Figure 1.** Flowchart of the calculation process.

Step 1. Formulate judgment matrix

To assess the BATT for waste LABs, the judgment matrix $A = \left(a_{ij}\right)_{n \times n}$ is established from expert grading following a 1–9 scale (see Table 2 and Equation (1)), where $a_{ij}$ represents the relative importance of $a_i$ and $a_j$ to criterion $C$, with constraints $a_{ij} \geq 0$ and $a_{ij} = \frac{1}{a_{ji}}$ $(i, j = 1, 2, \ldots, n)$.

$$A = \left(a_{ij}\right)_{n \times n} \begin{bmatrix} a_{11} & a_{12} & \cdots & a_{1n} \\ a_{21} & a_{22} & \cdots & a_{2n} \\ \vdots & \vdots & \vdots & \vdots \\ a_{n1} & a_{n2} & \cdots & a_{nn} \end{bmatrix}. \tag{1}$$

**Table 2.** Fundamental scale of absolute numbers.

| Relative Importance | Definition | Explanation |
|---|---|---|
| 1 | Equally important | Two attributes contribute equally to the target. |
| 3 | Slightly important | Experience and judgment lightly favor one attribute over another. |
| 5 | Strongly important | Experience and judgment strongly favor one attribute over another. |
| 7 | Very strongly important | An attribute is favored very strongly over another. |
| 9 | Absolutely important | The evidence favoring one attribute over another is of the highest possible order of affirmation. |
| 2, 4, 6 and 8 | Intermediate values between each two adjacent judgments | The importance is between the levels one point above and below. |

Step 2. Formulate attribute measure matrix

The attribute measure matrix $A = \left(u_{ij}\right)_{n \times n}$ is obtained with the transformation Equation (2) of the judgment matrix:

$$u_{ij} = \begin{cases} 0 & a_{ij} = 1,\ i = j \\ 0.5 & a_{ij} = 1,\ i \neq j \\ \frac{k}{k+1} & a_{ij} = k \\ \frac{1}{k+1} & a_{ij} = \frac{1}{k} \end{cases}, \tag{2}$$

where the matrix $A = \left(u_{ij}\right)_{n \times n}$ satisfies $u_{ij} \geq 0,\ u_{ji} \geq 0, u_{ij} + u_{ji} = 1, i \neq j$.

Step 3. Calculate relative weights of attributes

The relative weight $w_i$ is calculated from the Equation (3):

$$w_i = \frac{2}{n(n-1)} \sum_{j=1}^{n} u_{ij} \ \forall\ i = 1, 2, \ldots, n. \tag{3}$$

Thus, the weight vector is $w = (w_1,\ w_2,\ \ldots,\ w_n)$.

*3.2. Attribute Mathematical Theory*

Attribute mathematical theory can effectively solve the fuzzy multiple attribute decision problems [34–36]. Attribute measure is similar to membership degree in fuzzy mathematics. Compared with the fuzzy mathematical model, the sum of single index attribute measure and synthetic attribute measure is 1. According to the attribute mathematical theory, the corresponding attribute measure functions can be given for different attribute sets, and these functions are unique. Therefore, the classifications are more standardized and the evaluation results are more reliable [37]. The attribute comprehensive evaluation method uses attribute measure to determine which level the research object

belongs to and give a score. Evaluation levels are divided into excellent, good, medium, pass and poor, i.e., $T = \{t1, t2, t3, t4, t5\}$, respectively. The assessment steps are described below.

Step 1. Single index attribute measure analysis

The data format of the single index attribute measure $\mu_{xik}$ is shown in Table 3, where $a_{ik}(i = 1, 2, \cdots, m; and\ k = 0, 1, 2, \cdots, K)$ should meet $a_{i0} < a_{i1} < \ldots < a_{iK}$ or $a_{i0} > a_{i1} > \ldots > a_{iK}$. Considering that each indicator has different evaluation dimensions, it is necessary to unify the measurement of all indexes within an attribute evaluation system. $I_i$ represents the $i$ index, where the value range of $i$ is 1–$m$, which respectively indicates $m$ indicators of the BATT evaluation. $(C_1, C_2, \cdots, C_K)$ $(k = 0, 1, 2, \cdots, K)$ represents the evaluation sets of each indicator.

**Table 3.** Single index classification table.

| Evaluation Index | $C_1$ | $C_2$ | $\cdots$ | $C_K$ |
|---|---|---|---|---|
| $I_1$ | $a_{10} - a_{11}$ | $a_{11} - a_{12}$ | $\cdots$ | $a_{1K-1} - a_{1K}$ |
| $I_2$ | $a_{20} - a_{21}$ | $a_{21} - a_{22}$ | $\cdots$ | $a_{2K-1} - a_{2K}$ |
| $\cdots$ | $\cdots$ | $\cdots$ | $\cdots$ | $\cdots$ |
| $I_m$ | $a_{10} - a_{11}$ | $a_{11} - a_{12}$ | $\cdots$ | $a_{mK-1} - a_{mK}$ |

Assume the important two parameters ($b_{ik}$ and $d_{ik}$) that affect the attribute measure function are as follows:

$$b_{ik} = \frac{a_{ik-1} + a_{ik}}{2} \quad \forall\ k = 1, 2, \ldots, K, \tag{4}$$

$$d_{ik} = min\{|b_{ik} - a_{ik}|, |b_{ik+1} - a_{ik}|\} \quad \forall\ k = 1, 2, \ldots, K-1. \tag{5}$$

The single index attribute measure functions $\mu_{xik}(t)$ are as follows:

$$\mu_{xi1}(t) = \begin{cases} 1 & t < a_{i1} - d_{i1} \\ \frac{|t - a_{i1} - d_{i1}|}{2d_{i1}} & a_{i1} - d_{i1} \leq t \leq a_{i1} + d_{i1} \\ 0 & t > a_{i1} + d_{i1} \end{cases} \tag{6}$$

$$\mu_{xik}(t) = \begin{cases} 0 & t < a_{ik-1} - d_{ik-1} \\ \frac{|t - a_{ik-1} + d_{ik-1}|}{2d_{ik-1}} & a_{ik-1} - d_{ik-1} \leq t \leq a_{ik-1} + d_{ik-1} \\ 1 & a_{ik-1} + d_{ik-1} < t < a_{ik} - d_{ik} \\ \frac{|t - a_{ik} - d_{ik}|}{2d_{ik}} & a_{ik} - d_{ik} \leq p \leq a_{ik} + d_{ik} \\ 0 & t > a_{ik} + d_{ik} \end{cases} \tag{7}$$

$$\mu_{xiK}(t) = \begin{cases} 0 & t < a_{iK-1} - d_{iK-1} \\ \frac{|t - a_{iK-1} + d_{iK-1}|}{2d_{iK-1}} & a_{iK-1} - d_{iK-1} \leq t \leq a_{iK-1} + d_{iK-1}, \\ 1 & t > a_{iK-1} + d_{iK-1} \end{cases} \tag{8}$$

where $t$ is the value of each index; $k = 1, 2, \ldots, K-1$.

Step 2. Synthetic attribute measure analysis

Based on the single index attribute analysis and the result of the index weight, the synthetic attribute measure $u_{xk}$ can be expressed in Equation (9):

$$\mu_{xk} = \sum_{i=1}^{m} w_i \mu_{xik}, \tag{9}$$

where $\mu_{xk} \geq 0$, $\sum_{k=1}^{K} \mu_{xk} = 1$; $w_i$ is the weight of index $I_i$.

Step 3. Attribute recognition analysis

The aim of attribute recognition analysis is to select the BATT for waste LABs by the synthetic attribute measure $\mu_{xk}$ $(1 \le k \le K)$ and confidence criterion. The grade of the evaluation object is $C_{k_0}$, and $k_0$ is defined as:

If

$$k_0 = min\left\{k : \sum_{l=1}^{k} u_{xl} \ge \lambda, \quad 1 \le k \le K\right\}, \tag{10}$$

where $k = 1, 2, \ldots, K$; and $\lambda$ is the confidence coefficient, $\lambda = 0.6$–$0.7$.

### 3.3. Validation Experiments

In this study, the DST was selected as the verification technology according to the results of the BATT evaluation. The air pollution in the smelting process was controlled by gravity sedimentation, electrostatic precipitation, acid production, active carbon adsorption, and lime neutralization. Bag filter de-dusting and double-alkali desulfurization were used as air pollution control systems for other processes. The concentrations of Pb and $SO_2$ in the flue gas before and after end-of-pipe control (EPC) were detected online by flue gas analyzer. The discard slag was sampled based on the national standard sampling method (HJ/T 20-1998). The chemical components of the discard slag sample were checked by X-ray fluorescence (XRF).

## 4. Case Study

### 4.1. Sample Collection

Five waste LAB recycling technologies typically used in China were selected. Lead process flows of waste LAB treatments are illustrated in Figure 2. Process A refers to pre-desulfurization and multi-chamber smelting technology (PD-MCST). After automatic crushing and sorting, the lead grid, lead paste, plastic, battery separators, and electrolyte solutions are separately treated under closed conditions and negative pressure. The lead paste is desulfurized with $Na_2CO_3$, $(NH_4)_2CO_3$, NaOH or $Ca(OH)_2$, then smelted in a multi-chamber furnace using natural gas and pure oxygen as fuel. The lead grid is then smelted under low temperatures. Process B refers to pre-desulfurization and rotary kiln smelting technology (PD-RKST). The process flow is the same as for Process A, the biggest difference is that the smelting furnace is a rotary kiln. Process C refers to pre-desulfurization and blast furnace smelting technology (PD-BFST). In most cases, lead grid and lead paste are smelted together in this process. Process D refers to mixed smelting technology (MST). This is an improved technology for the mixed treatment of waste LABs and lead concentrate, but only based on primary lead smelting equipment. The biggest difference with process A is that this process does not require pre-desulfurization, but directly produces acid from flue gas to recover sulfur. Process E refers to direct smelting technology (DST). This is a novel process aimed at reducing the amount of hazardous waste and the cost. The biggest difference with Process A is that this process does not need pre-desulfurization, but directly produces acid from flue gas to recover sulfur. Compared with Process D, the biggest difference is that the smelting process does not require mixing with lead concentrate, and does not need to rely on primary lead smelting equipment.

Assuming that one ton of secondary lead was produced by these five processes, all relevant data are compared in Table 4. Most of the data used in this study were collected from the records of the enterprises located in Henan, Jiangsu, Hubei, Jiangxi, and Guizhou Provinces. Some data were obtained from public literature and interviews with environmental department professionals. The scope of sub-criteria in Table 4 was determined according to the current laws, regulations, industrial policies, and technical specifications of China, including the Law of the People's Republic of China on the Prevention and Control of Environmental Pollution by Solid Waste, Technical Policy for Pollution Prevention and Control in Lead-acid Battery Production and its Regeneration Process, Specification for

the Secondary Lead Industry, Clean Production Standard–Waste Lead Acid Battery Recycling Industry, Technical Specifications of Pollution Control for the Treatment of Lead-acid Batteries, and Emission Standards of Pollutants for the Secondary Copper, Aluminum, Lead and Zinc industry, etc.

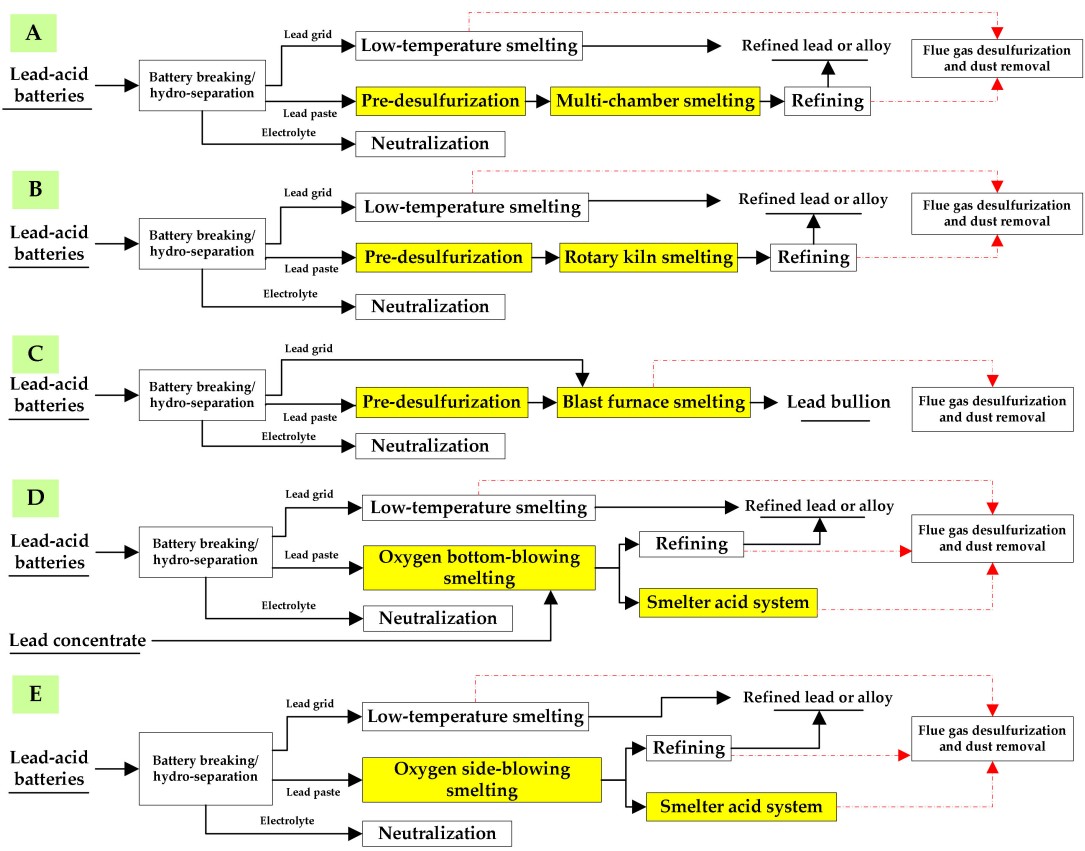

**Figure 2.** Lead process flow of waste LAB treatments.

**Table 4.** Representative data to be evaluated for the five technologies.

| Criteria | Sub-Criteria | Scope of Sub-Criteria | Unit | Process A | Process B | Process C | Process D | Process E |
|---|---|---|---|---|---|---|---|---|
| Environmental effect (A1) | Lead dust emission (A11) | 0–20 | g t$^{-1}$ | 2.06 | 8.53 | 10.88 | 4.90 | 3.33 |
| | Lead content of discard slag (A12) | 0–2 | % | 1.40 | 2.00 | 3.00 | 0.80 | 1.77 |
| | SO$_2$ emission (A13) | 0–1500 | g t$^{-1}$ | 421.40 | 490.59 | 1819.86 | 240.1 | 330.95 |
| | Discard slag (A14) | 0–150 | kg t$^{-1}$ | 148.00 | 115.00 | 269.00 | 102.00 | 133.00 |
| | NOx emission (A15) | 0–2000 | g t$^{-1}$ | 146.90 | 438.84 | 1176.00 | 208.84 | 217.66 |
| Comprehensive resource utilization (A2) | Utilization of lead (A21) | 98–100 | % | 99.80 | 98.50 | 98.00 | 99.70 | 99.20 |
| | Utilization of sulfur (A22) | 95–100 | % | 98.00 | 96.00 | 95.60 | 98.00 | 98.88 |
| | Disposal of electrolyte (A23) | 90–100 | % | 100.00 | 100.00 | 100.00 | 100.00 | 100.00 |
| | Disposal of discard slag (A24) | 0–100 | % | 100.00 | 100.00 | 100.00 | 100.00 | 100.00 |
| | Utilization of plastic (A25) | 95–100 | % | 100.00 | 98.48 | 98.00 | 100.00 | 100.00 |
| | Utilization of wastewater (A26) | 90–100 | % | 100.00 | 100.00 | 100.00 | 100.00 | 100.00 |
| Technical performance (A3) | Industrial policies (A31) | 0, 1 | — | 1 | 1 | 1 | 1 | 1 |
| | Smelting process and equipment (A32) [9,25] | Level 1, 2, 3 | — | Level 1 | Level 2 | Level 3 | Level 1 | Level 1 |
| | State of technology reliability (A33) [25] | Level 1, 2, 3 | — | Level 1 | Level 1 | Level 1 | Level 1 | Level 1 |
| | Automation level (A34) | Level 1, 2, 3 | — | Level 1 | Level 2 | Level 3 | Level 1 | Level 1 |
| | State of disposal site (A35) | Level 1, 2, 3 | — | Level 1 | Level 1 | Level 3 | Level 1 | Level 1 |
| Material and energy consumption (A4) | Comprehensive energy consumption (A41) | 100–130 | kgce t$^{-1}$ | 94.50 | 109.60 | 130.00 | 97.80 | 95.18 |
| | Fresh water consumption (A42) | 0.1–0.5 | m$^3$ t$^{-1}$ | 0.21 | 0.27 | 0.35 | 0.56 | 1.21 |
| | Auxiliary materials consumption (A43) | 0.1–0.3 | t t$^{-1}$ | 0.20 | 0.24 | 0.22 | 0.20 | 0.11 |
| Economic performance (A5) | Investment profit (A51) | 10–20 | % | 15.06 | 14.50 | 32.50 | 16.00 | 15.40 |
| | Operational cost (A52) | 341.3–455.0 | USD t$^{-1}$ | 426.6 | 398.2 | 321.4 | 355.5 | 376.8 |
| Environmental management (A6) | Environmental laws and regulations (A61) | 0, 1 | — | 1 | 1 | 1 | 1 | 1 |
| | Environmental management system and organization (A62) | Level 1, 2, 3 | — | Level 1 | Level 1 | level 2 | Level 1 | Level 1 |
| | Environmental emergency (A63) | Level 1, 2, 3 | — | Level 1 | Level 1 | level 2 | Level 1 | Level 1 |

### 4.2. Weights Analysis of Criteria and Sub-Criteria

In this study, the criteria affecting the evaluation of the BATT for waste LABs was defined as: A1 = environment effect, A2 = comprehensive resource utilization, A3 = technical performance, A4 = material and energy consumption, A5 = economic performance, A6 = environmental management. The relative weights of each indicator were determined via the Delphi-AHM method. First, we formulated the judgment matrix. Then, experts from environmental departments, enterprises, and scientific research institutes compared the elements in the judgment matrix, using a 1–9 scale. Finally, the relative weights of each indicator were calculated by the AHM. The judgment matrix was:

$$
\begin{bmatrix}
1 & 2 & 3 & 5 & 7 & 5 \\
1/2 & 1 & 3 & 2 & 6 & 3 \\
1/3 & 1/3 & 1 & 2 & 4 & 4 \\
1/5 & 1/2 & 1/2 & 1 & 2 & 2 \\
1/7 & 1/6 & 1/4 & 1/2 & 1 & 2 \\
1/5 & 1/3 & 1/4 & 1/2 & 1/2 & 1
\end{bmatrix},
$$

which was transformed to the measure matrix

$$
\begin{bmatrix}
0 & 0.67 & 0.75 & 0.83 & 0.88 & 0.83 \\
0.33 & 0 & 0.75 & 0.67 & 0.86 & 0.75 \\
0.25 & 0.25 & 0 & 0.67 & 0.80 & 0.80 \\
0.17 & 0.33 & 0.33 & 0 & 0.67 & 0.67 \\
0.12 & 0.14 & 0.20 & 0.33 & 0 & 0.67 \\
0.17 & 0.25 & 0.20 & 0.33 & 0.33 & 0
\end{bmatrix},
$$

and the weights of criteria indicators were $w = (0.264, 0.224, 0.184, 0.145, 0.098, 0.085)$. The results indicated that the environmental effect was identified as the most important aspect, followed by comprehensive resource utilization, technical performance, material and energy consumption, economic performance, and environmental management.

Weights for the secondary level indicators of the criterion layer, i.e., A1, A2, A3, A4, A5, A6, were obtained similarly, and the corresponding measure matrix was

$$
\begin{bmatrix}
0 & 0.67 & 0.80 & 0.83 & 0.80 \\
0.33 & 0 & 0.67 & 0.67 & 0.67 \\
0.20 & 0.33 & 0 & 0.67 & 0.67 \\
0.17 & 0.33 & 0.33 & 0 & 0.67 \\
0.20 & 0.33 & 0.33 & 0.33 & 0
\end{bmatrix}
$$

$$
\begin{bmatrix}
0 & 0.80 & 0.75 & 0.75 & 0.80 & 0.80 \\
0.20 & 0 & 0.67 & 0.67 & 0.75 & 0.67 \\
0.25 & 0.33 & 0 & 0.67 & 0.75 & 0.67 \\
0.25 & 0.33 & 0.33 & 0 & 0.75 & 0.67 \\
0.20 & 0.25 & 0.25 & 0.25 & 0 & 0.67 \\
0.20 & 0.33 & 0.33 & 0.33 & 0.33 & 0
\end{bmatrix}
$$

$$
\begin{bmatrix}
0 & 0.67 & 0.80 & 0.80 & 0.75 \\
0.33 & 0 & 0.80 & 0.80 & 0.80 \\
0.20 & 0.20 & 0 & 0.50 & 0.67 \\
0.20 & 0.20 & 0.50 & 0 & 0.67 \\
0.25 & 0.20 & 0.33 & 0.33 & 0
\end{bmatrix}
$$

$$
\begin{bmatrix}
0 & 0.75 & 0.80 \\
0.25 & 0 & 0.67 \\
0.20 & 0.33 & 0
\end{bmatrix}
$$

$$
\begin{bmatrix}
0 & 0.67 \\
0.33 & 0
\end{bmatrix}
$$

and

$$
\begin{bmatrix}
0 & 0.67 & 0.67 \\
0.33 & 0 & 0.67 \\
0.33 & 0.33 & 0
\end{bmatrix}.
$$

The corresponding weights of sub-criteria were $w(A11, A12, A13, A14, A15) = (0.310, 0.233, 0.187, 0.150, 0.120)$, $w(A21, A22, A23, A24, A25, A26) = (0.260, 0.197, 0.178, 0.155, 0.108, 0.102)$, $w(A31, A32, A33, A34, A35) = (0.302, 0.273, 0.157, 0.157, 0.111)$, $w(A41, A42, A43) = (0.517, 0.305, 0.178)$ and $w(A51, A52) = (0.667, 0.333)$, $w(A61, A62, A63) = (0.447, 0.333, 0.220)$, respectively.

The global weights of the sub-criteria were calculated based on Equation (3), as shown in Figure 3. The results indicated that lead dust emission had a global weight of 0.082 among the 24 examined indicators, and was therefore the most important indicator for BATT evaluation.

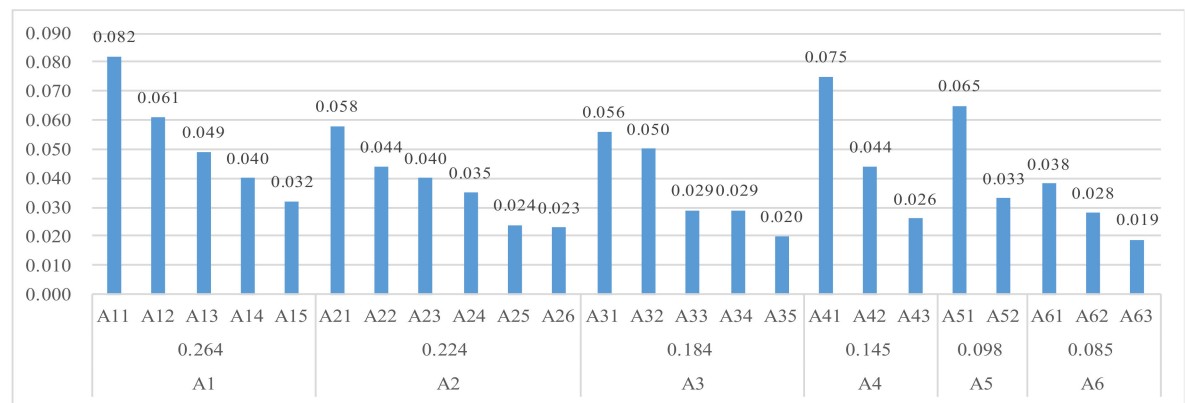

**Figure 3.** Global weights of the sub-criteria.

### 4.3. Evaluation of the BATT for Waste LABs

The evaluation system of the BATT (Table 1) was applied in this stage, and the evaluation steps can be carried out as follows:

(1)  According to the classification of the single index (Table 3) and the data presented in Table 4, attribute measure functions of evaluation indicators, e.g., $I_{A_{11}}$, $I_{A_{12}}$, $I_{A_{13}}$, $I_{A_{14}}$ and $I_{A_{15}}$, can be obtained by using Equations (4)–(8), as shown in Figure 4. The functions of other indicators, i.e., $I_{A_{21}}$, $I_{A_{22}}$, …, $I_{A_{63}}$, are obtained similarly. The calculated attribute measure values of sub-criteria single indicators are shown in Table 5.

(2)  The synthetic attribute measures of the five processes can be computed by using Equation (9) and the computed results are shown in Table 6.

(3)  Based on the obtained synthetic attribute measures, the pollution control grade of each technology can be determined by using Equation (10). Considering the significant impact of environment pollution, λ was taken as 0.7 in the evaluation, so that the selected technologies can minimize environmental pollution to the maximum extent. Then in the evaluation stage, Equation (10) can be written as

$$
k_0 = min\left\{ k : \sum_{l=k}^{5} u_{xl} \geq 0.7, \quad 1 \leq k \leq 5 \right\}. \tag{11}
$$

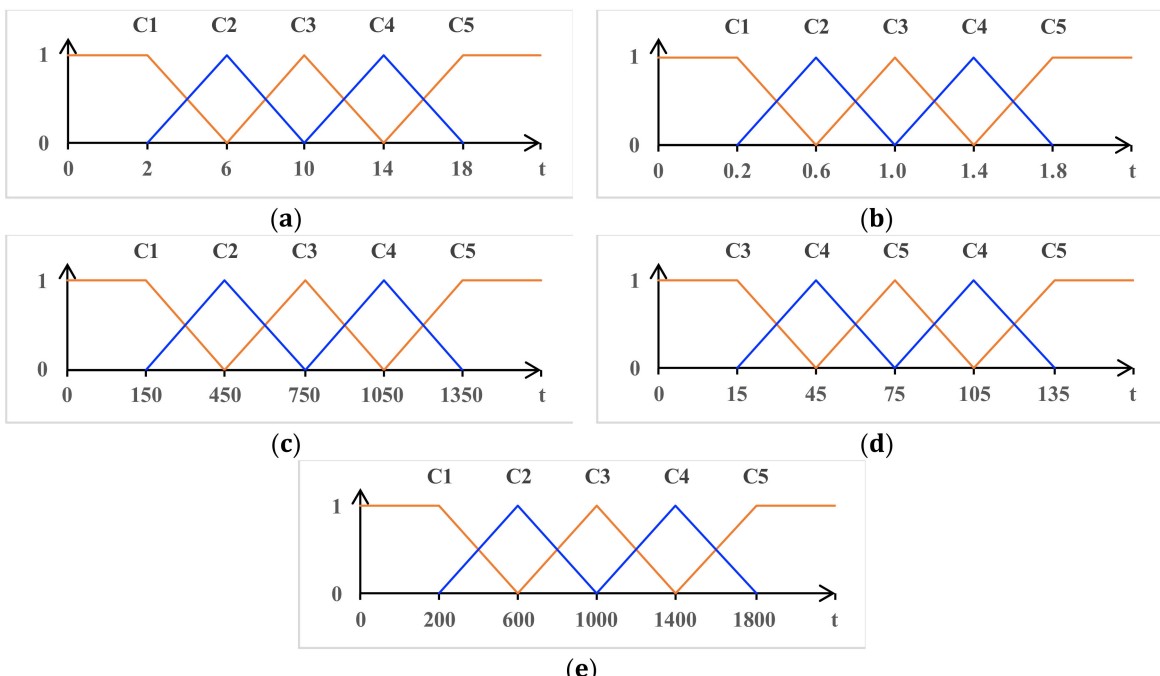

**Figure 4.** Attribute measure functions of evaluation indicators. (**a**) $I_{A_{11}}$, (**b**) $I_{A_{12}}$, (**c**) $I_{A_{13}}$, (**d**) $I_{A_{14}}$, (**e**) $I_{A_{15}}$.

**Table 5.** The attribute measure values of sub-criteria single indicators.

| Sub-Criteria | Process A {$C_1,C_2,C_3,C_4,C_5$,} | Process B {$C_1,C_2,C_3,C_4,C_5$,} | Process C {$C_1,C_2,C_3,C_4,C_5$,} | Process D {$C_1,C_2,C_3,C_4,C_5$,} | Process E {$C_1,C_2,C_3,C_4,C_5$,} |
|---|---|---|---|---|---|
| A11 | {0.99, 0.01, 0, 0, 0} | {0, 0.37, 0.63, 0, 0} | {0, 0, 0.78, 0.22, 0} | {0.28, 0.72, 0, 0, 0} | {0.67, 0.33, 0, 0, 0} |
| A12 | {0, 0, 0, 1, 0} | {0, 0, 0, 0, 1} | {0, 0, 0, 0, 1} | {0, 0.5, 0.5, 0, 0} | {0, 0, 0, 0.08, 0.92} |
| A13 | {0.1, 0.9, 0, 0, 0} | {0, 0.86, 0.14, 0, 0} | {0, 0, 0, 0, 1} | {0.7, 0.3, 0, 0, 0} | {0.4, 0.6, 0, 0, 0} |
| A14 | {0, 0, 0, 0, 1} | {0, 0, 0, 0.67, 0.33} | {0, 0, 0, 0, 1} | {0, 0, 0.1, 0.9, 0} | {0, 0, 0, 0.07, 0.93} |
| A15 | {1, 0, 0, 0, 0} | {0.4, 0.6, 0, 0, 0} | {0, 0, 0.56, 0.44, 0} | {0.98, 0.02, 0, 0, 0} | {0.96, 0.04, 0, 0, 0} |
| A21 | {1, 0, 0, 0, 0} | {0, 0, 0, 0.75, 0.25} | {0, 0, 0, 0, 1} | {0.75, 0.25, 0, 0, 0} | {0, 0.5, 0.5, 0, 0} |
| A22 | {0, 0.5, 0.5, 0, 0} | {0, 0, 0, 0.5, 0.5} | {0, 0, 0, 0.1, 0.9} | {0, 0.5, 0.5, 0, 0} | {0.38, 0.62, 0, 0, 0} |
| A23 | {1, 0, 0, 0, 0} | {1, 0, 0, 0, 0} | {1, 0, 0, 0, 0} | {1, 0, 0, 0, 0} | {1, 0, 0, 0, 0} |
| A24 | {1, 0, 0, 0, 0} | {1, 0, 0, 0, 0} | {1, 0, 0, 0, 0} | {1, 0, 0, 0, 0} | {1, 0, 0, 0, 0} |
| A25 | {1, 0, 0, 0, 0} | {0, 0.98, 0.02, 0, 0} | {0, 0.5, 0.5, 0, 0} | {1, 0, 0, 0, 0} | {1, 0, 0, 0, 0} |
| A26 | {1, 0, 0, 0, 0} | {1, 0, 0, 0, 0} | {1, 0, 0, 0, 0} | {1, 0, 0, 0, 0} | {1, 0, 0, 0, 0} |
| A31 | {1, 0, 0, 0, 0} | {1, 0, 0, 0, 0} | {0, 0, 0, 0, 1} | {1, 0, 0, 0, 0} | {1, 0, 0, 0, 0} |
| A32 | {1, 0, 0, 0, 0} | {1, 0, 0, 0, 0} | {1, 0, 0, 0, 0} | {1, 0, 0, 0, 0} | {1, 0, 0, 0, 0} |
| A33 | {1, 0, 0, 0, 0} | {0, 0, 1, 0, 0} | {0, 0, 0, 0, 1} | {1, 0, 0, 0, 0} | {1, 0, 0, 0, 0} |
| A34 | {1, 0, 0, 0, 0} | {0, 0, 1, 0, 0} | {0, 0, 0, 0, 1} | {1, 0, 0, 0, 0} | {1, 0, 0, 0, 0} |
| A35 | {1, 0, 0, 0, 0} | {1, 0, 0, 0, 0} | {0, 0, 0, 0, 1} | {1, 0, 0, 0, 0} | {1, 0, 0, 0, 0} |
| A41 | {1, 0, 0, 0, 0} | {0, 0.9, 0.1, 0, 0} | {0, 0, 0, 0, 1} | {1, 0, 0, 0, 0} | {1, 0, 0, 0, 0} |
| A42 | {0.13, 0.87, 0, 0, 0} | {0, 0.37, 0.63, 0, 0} | {0, 0, 0.38, 0.62, 0} | {0, 0, 0, 0, 1} | {0, 0, 0, 0, 1} |
| A43 | {0, 0, 1, 0, 0} | {0, 0, 0, 1, 0} | {0, 0, 0.5, 0.5, 0} | {0, 0, 1, 0, 0} | {1, 0, 0, 0, 0} |
| A51 | {0, 0.03, 0.97, 0, 0} | {0, 0.25, 0.75, 0, 0} | {1, 0, 0, 0, 0} | {0, 0.5, 0.5, 0, 0} | {0, 0.2, 0.8, 0, 0} |
| A52 | {0, 0, 0, 0.75, 0.25} | {0, 0, 1, 0, 0} | {1, 0, 0, 0, 0} | {0.88, 0.12, 0, 0, 0} | {0, 0.94, 0.06, 0, 0} |
| A61 | {1, 0, 0, 0, 0} | {1, 0, 0, 0, 0} | {1, 0, 0, 0, 0} | {1, 0, 0, 0, 0} | {1, 0, 0, 0, 0} |
| A62 | {1, 0, 0, 0, 0} | {1, 0, 0, 0, 0} | {0, 0, 1, 0, 0} | {1, 0, 0, 0, 0} | {1, 0, 0, 0, 0} |
| A63 | {1, 0, 0, 0, 0} | {1, 0, 0, 0, 0} | {0, 0, 1, 0, 0} | {1, 0, 0, 0, 0} | {1, 0, 0, 0, 0} |

**Table 6.** Evaluation results of the BATT for waste LABs.

|  | $u_{xk}$ | Level | Rank |
|---|---|---|---|
| Process A | {0.6478, 0.1076, 0.1111, 0.0858, 0.0482} | Good | 2 |
| Process B | {0.3218, 0.2157, 0.2340, 0.1183, 0.1107} | Medium | 4 |
| Process C | {0.2840, 0.0120, 0.1706, 0.0768, 0.4566} | Poor | 5 |
| Process D | {0.6272, 0.1787, 0.1150, 0.0360, 0.0440} | Good | 1 |
| Process E | {0.6140, 0.1580, 0.0830, 0.0077, 0.1379} | Good | 3 |

The results indicated that Processes D and C were respectively ranked as the best and worst technologies from the five commonly used technologies examined herein. Processes A and E ranked slightly lower than Process D, but they were also at a Good level, whereas Process C was found to be at a Medium level.

*4.4. Analysis of Overall Evaluation Results*

The detailed comparison chart of the five technologies in criteria and sub-criteria layers is illustrated in Figure 5. (1) It can be seen that Process C consumed considerable amounts of energy and generated the most pollutants; therefore, this technology should be deemed obsolete. Moreover, despite the remarkable economic performance of this technology, practices that prioritize economy over environmental and energy conservation should be considered unsustainable. Based on the evaluation results, Process C barely met the current environmental requirements, and although this technology is not yet expressly prohibited, it is reportedly nearly no longer approved in China. (2) Although Process B implemented pre-desulfurization, the indexes of A11, A12, A21, A22, A33, and A41 were still lower than those of Process A. This was because rotary kiln smelting is classified as intermittent smelting, which results in high-energy consumption, high lead content in the discard slag, and low lead and sulfur utilization [38]. Considering the lead and sulfur utilization as well as terminal pollutant discharge, it was concluded that less investment was required for environmental protection and technology improvement, thereby leading to a slight economic advantage. (3) Process D assumed the leading position in all indicators except A11, A22, A42, and A43. That was because this process employed flue gas to make acid instead of implementing pre-desulfurization, which improves smelting temperature, leads to direct freshwater consumption, and emits slightly more lead dust than Process A, which was the same case for Process E. Li et al. also found that the lead dust emission of Process D was greater than that of Process A [11]. (4) Process A performed slightly better in terms of material and energy consumption and slightly worse in terms of economic performance, compared with Processes D and E. This was because Process A adopted automatic and continuous closed feeding to ensure the exchange of heat energy and the integration of material preheating, melting, and smelting [25]. However, it produced a high amount of $Na_2SO_4$ during the pre-desulfurization process, which has little economic value and is identified by the Identification Standards for Hazardous Waste General Rules (GB 5085.7-2019) [39]. (5) Process E was similar to Process A in terms of environmental effect, comprehensive resource utilization, technical performance, and environmental management. Because it solved the problems of high pre-desulfurization cost and by-product treatment, this process was found to have certain economic advantages [40]. Nevertheless, due to its short development time, the lead content of discard slag and freshwater consumption should be further controlled.

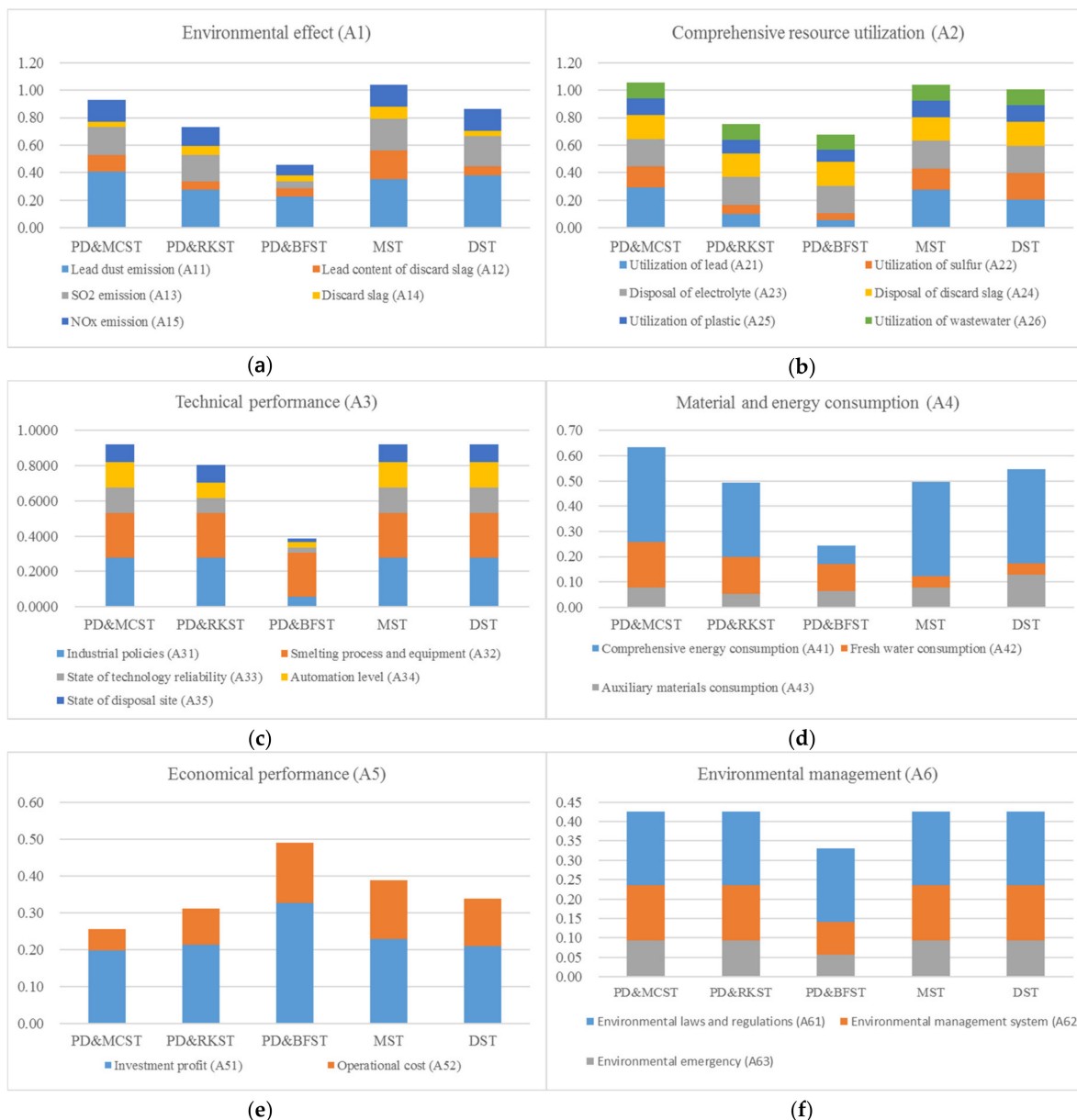

**Figure 5.** Detailed comparison chart of the criteria and sub-criteria of the five technologies. (**a**) Environmental performance (A11–A15); (**b**) Comprehensive resource utilization (A21–A26); (**c**) Technical performance (A31–A35); (**d**) Material and energy consumption (A41–A43); (**e**) Economic performance (A51–A52); (**f**) Environmental management (A61–A63).

Each technology has its limitations, and the above-described technologies are no exception. Based on our comprehensive evaluation results, we concluded that Processes D, A, and E should be recommended as BATTs for waste LABs at this stage. In the long-term future, when the proportion of waste LABs increases to a certain level, we should reduce the further promotion of Process D because it relies on the primary lead smelting equipment. Moreover, minimizing waste generation is critical in the process of becoming a cleaner and more competitive nation. Process optimization is therefore a necessary measure to reduce the generation of solid waste, which represents a substantial challenge for Process A. Overall, Process E has considerable economic benefits and low environmental effect, produces low amounts of solid waste, and does not need to rely on primary lead smelting equipment, and therefore has great potential for future development.



### 4.5. Validation Study on the Feasibility of BATT

The DST was selected as an example to preliminarily verify the pollution control level of the BATT recommended for waste LABs. As shown in Figure 6a,b, the concentration of Pb and $SO_2$ in side-blowing smelting was the highest, reaching 15,030 mg·m$^{-3}$ and 95,611 mg·m$^{-3}$, respectively. Their concentrations were then reduced to 0.75 mg·m$^{-3}$ and 119 mg·m$^{-3}$, respectively, by the combined EPC treatment of gravity sedimentation, electrostatic precipitation, acid production, active carbon adsorption, and lime neutralization. The final emissions of Pb and $SO_2$ were 3.72 g·t$^{-1}$ and 354.5 g·t$^{-1}$, respectively, which were far lower than the requirements of the emission standard of China (Pb 20 g·t$^{-1}$; $SO_2$ 1500 g·t$^{-1}$) (Figure 7) [41]. XRF results showed that the Pb concentration in the discard slag was 1.80 wt%, which met the requirements of the emission standard of China (2 wt%) [42]. The results showed that the DST can meet the requirements of pollution control, which is consistent with the evaluation results.

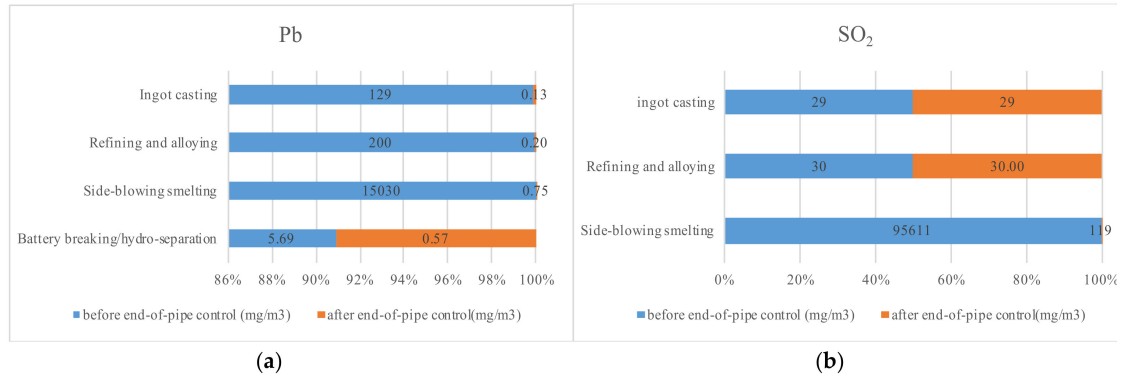

(**a**) (**b**)

**Figure 6.** Pb/$SO_2$ in flue gas before and after end-of-pipe treatment. (**a**) Pb (**b**) $SO_2$.

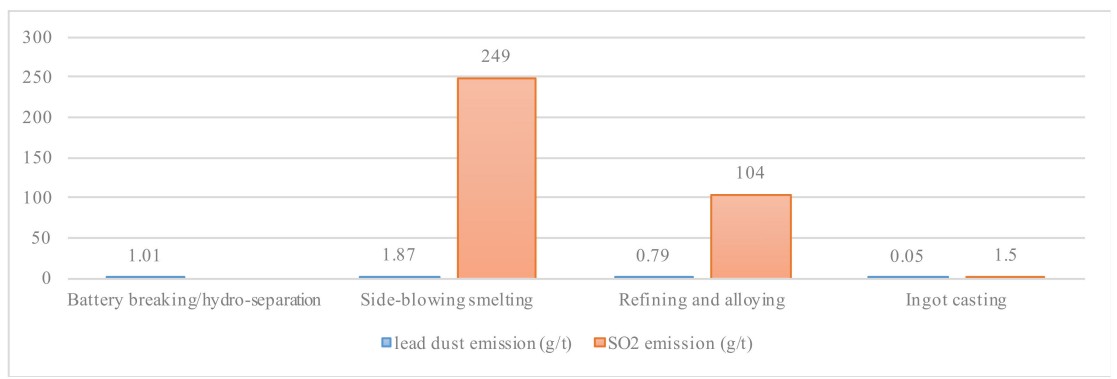

**Figure 7.** The Emissions of Pb and $SO_2$ after end-of-pipe treatment.

## 5. Conclusions

The aim of this research was to evaluate the BATT for waste LABs. To this end, 24 effective indicators were identified by literature review and expert consultation. Then, the weight of each indicator was determined by the Delphi-AHM method. According to our assessment, the environmental effect dimension was identified as the most important aspect, followed by comprehensive resource utilization, technical performance, material and energy consumption, economic performance, and environmental management, in descending order. The environmental effect and comprehensive resource utilization were of significant importance because resource utilization and pollutant emission are the core criteria to determine whether a technology is the BATT for a given pollutant. For sub-criteria, (1) The lead dust emission index was identified as the most important factor in the environmental effect dimension, followed by lead content of discard slag, $SO_2$ emission, discard slag, and NOx emission, in descending order. This weight distribution may be due to the fact that both the existing secondary lead industry

and newly established factories are forbidden from discharging lead-containing wastewater, waste electrolyte solutions, or high lead slag during the production process. The discharged lead and sulfur mainly enter the environment through flue gas and propagate with air diffusion, thereby affecting air quality [11]. The soil is polluted by dry settlement and wet settlement, and a small amount of lead also exists in smelting residue. Moreover, a large amount of smelting residue is typically produced in the smelting process, which is a common industry problem. Direct landfill disposal of these waste materials does not only lead to resource waste but also increases costs. Therefore, solid waste reduction can be achieved if the Pb content index is strictly controlled. (2) The utilization of lead and utilization of sulfur indices occupied a higher weight in the comprehensive resource utilization dimension, compared with other indicators. That may be because the waste acid, wastewater, flue gas, and solid waste generated during the waste LAB treatment process contained both lead and sulfur. Importantly, serious environmental impacts are likely to occur if this recovery rate cannot be improved. (3) The industrial policies index was presented as the most influential technical performance factor. This index is the minimum requirement of environmental supervision, and therefore legal enterprises should at least meet this requirement. The state of disposal site index had the lowest significance, mainly due to the fewer requirements of policies and standards associated with this index. Moreover, state of disposal site is not a key indicator of environmental monitoring, and only attracts people's attention in the event of an accident. (4) Compared with other factors in the material and energy consumption dimension, one of the reasons why the comprehensive energy consumption index exhibited a higher weight may be that this index reflects the total energy consumption during the production processing. The smelting industry must therefore focus on green development strategies (e.g., saving energy and reducing consumption) to continue thriving in an increasingly complex and fiercely competitive context. (5) The investment profit index was chosen as the main index for economic evaluation, and reflects the amount of estimated profit associated with technology investment. The reason why this index was highly ranked may be that technology must generate profits to be valuable. If profits do not increase, no amount of turnover will help, and investors tend to stay away from enterprises with low profit rates in favor of higher profits. (6) Environmental management came last in our assessment. This does not mean that environmental management is unimportant. Environmental management includes mainly various system requirements related to pollution control. But the actual situation shows that not all high standard requirements can be strictly implemented. Moreover, environmental effect, material and energy consumption, and comprehensive resource utilization fully reflect the implementation effect of environmental management, so it is more important.

Eventually, the attribute measure functions were constructed to calculate single index attribute and synthetic attribute, and a confidence criterion was selected to recognize the grade of the studied technologies. The evaluation results showed that MST, PD-MCST, and DST should be recommended as the BATT for waste LABs at this stage, while in the long-term future, DST has great potential for future development. Thus, the DST was selected as the verification technology, according to the evaluation results. The results indicated that the DST can meet the requirements of pollution control, which is consistent with the evaluation results.

The indicators proposed herein have several implications for scholars and government decision-makers. The evaluation results can be used to guide enterprises in selecting optimal technologies and provide technical support for the improvement of the technological level and environmental protection efficiency of the secondary lead industry.

Similar to other methods, the proposed method has its limitations. Not all evaluation indicators are classified quantitatively, and some of them were obtained via the evaluation of experts, which introduces a certain degree of subjectivity.

**Author Contributions:** Writing—original draft: W.W.; investigation: Y.H.; validation: D.Z.; methodology: Y.W.; data curation, D.P. All authors have read and agreed to the published version of the manuscript.

**Funding:** This research was financially supported by the National Key R&D Program of China (Grant No. 2018YFC1902803), and the Beijing Social Science Fund (No. 17YJA001).

**Acknowledgments:** We would like to thank environmental and non-ferrous metallurgy experts and the work of secondary lead enterprises for their help with the research.

**Conflicts of Interest:** The authors declare no conflict of interest.

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
