# Peer review of "Multi-Criteria Evaluation of Best Available Treatment Technology for Waste Lead-Acid Battery: The Case of China"

_sustainability, doi:10.3390/su12114479_

Round 1

Reviewer 1 Report

Dear authors, I have reviewed your manuscript and it is well written and good to read.

I have some comments:

General: Lead acid battery recycling was matured in Europe in the last century and this might impact the imbalance of Chinese articles to publications to other contintents. Maybe you can check also older publications. Of course this is a topic gaining high interest in China over the last 10 years and this is the reason for the high amount of Chinese publications. SInce your title says, that this is a case study of CHina I do not see a problem here.

Also your selected indicators are unclear. E.g. what is "industrial policy" under technical performance. Maybe you provide supplementary information where the indicators are explained.

In Detail:

figure 1: I am a process engineer and therefore I am not ok with the figure 1 and the heading: "Complete" process flow. This is incorrect as the whole acid treatment is missing here. Maybe you want to change the title in "Lead process flow of waste LAB treatment". As the acid treatment is often the backbone of the feasibility of the recycling plant this must be clarified. Please see also https://www.ncbi.nlm.nih.gov/pmc/articles/PMC5990833/

Maybe it would help to describe the standard lead acid battery in CHina (composition). Where does the H2SO4 go to in your process? Nowbody wants that high amount of sulfur in a furnace. Normally the acid is neutralised and goes then further to the Chemistry industry. Please also check: http://archive.basel.int/pub/techguid/tech-wasteacid.pdf and  https://apps.who.int/iris/bitstream/handle/10665/259447/9789241512855-eng.pdf

CHapter 4: It is still unclear if the numbers in table are the measured data or the "allowed" data. You cite in line 261 that the numbers are reported to the province. How does it work in CHina. Does it mean that every citizen can access this data? Or how did you get it? Please add also a year/month or even date to the table 4. And please explain level 1-3

Figure 4 is not easy to understand without a legend. YOu have to read the paper 3 times to understand what number belongs to which process and what is the difference. Please insert also Process names , as ABCDE was mentioned very early in the paper and you have always to scroll backwards. Why is there no achsis description?

COnclusion: is very short and I am missing more critical reflection of this method. Can you clearly sort numbers for eg lead dust (that can kill people or challenge disabled babies) compare to the economic performance? I personally have a problem with it. For example your indicator "environmental management" came last in your assessment. What does it mean? The pollution of soil and water is less important than anything else in the assessment?

That is a bit too easy for me and does not reflect our approach to protect the planet's ressources.

Author Response

Dear Reviewer #1:

    Thank you for your letter and for the reviewers’ comments concerning our manuscript entitled Multi-criteria evaluation of best available treatment technology for waste lead acid battery: the case of China (ID: sustainability-810633). Those comments are all valuable and very helpful for revising and improving our paper, as well as an important guide significant to our research. We have studied comments carefully and have made correction which we hope meet with approval. Revised portion are marked in red in the paper.

    The main corrections in the paper and the response to the reviewer’s comments are as attached.

Reviewer 2 Report

The reviewer has the following comments:

In line 174, after the introduction and the literature review, is the beginning of your contribution, i.e. the suggested methodology. At that place you may want clearly to explain the aim and the novelty in your research.

You describe the calculation process in steps (parts 3.1. and 3.3.) which makes them clear, but usually such an approach is more applicable for iterative procedures. Do you need iterations or the algorithm is linear? A chart diagram would be helpful.

You may want to extend the part 3.3. Validation experiments. The experiments are not clearly presented.

Author Response

Dear Reviewer #2:

    Thank you for your letter and for the reviewers’ comments concerning our manuscript entitled Multi-criteria evaluation of best available treatment technology for waste lead acid battery: the case of China (ID: sustainability-810633). Those comments are all valuable and very helpful for revising and improving our paper, as well as an important guide significant to our research. We have studied comments carefully and have made correction which we hope meet with approval. Revised portion are marked in red in the paper.

    The main corrections in the paper and the response to the reviewer’s comments are as attached.

Reviewer 3 Report

1.For section 4 of 'case study', the authors are required to provide the sources/references (links) of the results used in table 4. Besides, the authors are suggested to present the results in the statistical format of 'average' along with 'deviation' to illustrate the reproducibility/repeatability. 

2.The authors are recommended to provide typical XRF profiles of discard slag sample discussed in section 3.3.

3.For section 4.2, the authors are required to provide detailed criteria for the weighting factor assignments? It is not scientifically rigorous to make a statement that "their relative importance was determined by expert experience..."

Author Response

Dear Reviewer #3:

    Thank you for your letter and for the reviewers’ comments concerning our manuscript entitled Multi-criteria evaluation of best available treatment technology for waste lead acid battery: the case of China (ID: sustainability-810633). Those comments are all valuable and very helpful for revising and improving our paper, as well as an important guide significant to our research. We have studied comments carefully and have made correction which we hope meet with approval. Revised portion are marked in red in the paper.

    The main corrections in the paper and the response to the reviewer’s comments are as attached.
